# Discovery Viewer (DV): Web-Based Medical AI Model Development Platform and Deployment Hub

**DOI:** 10.3390/bioengineering10121396

**Published:** 2023-12-06

**Authors:** Valentin Fauveau, Sean Sun, Zelong Liu, Xueyan Mei, James Grant, Mikey Sullivan, Hayit Greenspan, Li Feng, Zahi A. Fayad

**Affiliations:** 1BioMedical Engineering and Imaging Institute (BMEII), Icahn School of Medicine at Mount Sinai, New York, NY 10029, USA; zelong.liu@icahn.mssm.edu (Z.L.); xueyan.mei@icahn.mssm.edu (X.M.); james.grant@mssm.edu (J.G.); mikey.sullivan@mssm.edu (M.S.); hayit.greenspan@mssm.edu (H.G.); zahi.fayad@mssm.edu (Z.A.F.); 2Department of Diagnostic, Molecular, and Interventional Radiology, Icahn School of Medicine at Mount Sinai, New York, NY 10029, USA; sean.sun@mountsinai.org; 3Center for Advanced Imaging Innovation and Research (CAI^2^R), NYU Grossman School of Medicine, New York, NY 10016, USA; li.feng@nyulangone.org

**Keywords:** AI, medicine, Web app, imaging, medical viewers, transfer learning, federated learning, digital twin

## Abstract

The rapid rise of artificial intelligence (AI) in medicine in the last few years highlights the importance of developing bigger and better systems for data and model sharing. However, the presence of Protected Health Information (PHI) in medical data poses a challenge when it comes to sharing. One potential solution to mitigate the risk of PHI breaches is to exclusively share pre-trained models developed using private datasets. Despite the availability of these pre-trained networks, there remains a need for an adaptable environment to test and fine-tune specific models tailored for clinical tasks. This environment should be open for peer testing, feedback, and continuous model refinement, allowing dynamic model updates that are especially important in the medical field, where diseases and scanning techniques evolve rapidly. In this context, the Discovery Viewer (DV) platform was developed in-house at the Biomedical Engineering and Imaging Institute at Mount Sinai (BMEII) to facilitate the creation and distribution of cutting-edge medical AI models that remain accessible after their development. The all-in-one platform offers a unique environment for non-AI experts to learn, develop, and share their own deep learning (DL) concepts. This paper presents various use cases of the platform, with its primary goal being to demonstrate how DV holds the potential to empower individuals without expertise in AI to create high-performing DL models. We tasked three non-AI experts to develop different musculoskeletal AI projects that encompassed segmentation, regression, and classification tasks. In each project, 80% of the samples were provided with a subset of these samples annotated to aid the volunteers in understanding the expected annotation task. Subsequently, they were responsible for annotating the remaining samples and training their models through the platform’s “Training Module”. The resulting models were then tested on the separate 20% hold-off dataset to assess their performance. The classification model achieved an accuracy of 0.94, a sensitivity of 0.92, and a specificity of 1. The regression model yielded a mean absolute error of 14.27 pixels. And the segmentation model attained a Dice Score of 0.93, with a sensitivity of 0.9 and a specificity of 0.99. This initiative seeks to broaden the community of medical AI model developers and democratize the access of this technology to all stakeholders. The ultimate goal is to facilitate the transition of medical AI models from research to clinical settings.

## 1. Introduction

The rapid advancement of AI technologies has made important breakthroughs in various sectors, including healthcare, where great progress has been made with applications ranging from diagnostics, prognostics, medical imaging segmentations, and beyond. Nevertheless, the existing approach for data and model sharing remains limited, and many of the developed medical models remain static and unused after publication. The current dynamic diverges from what we now understand should be the essence of applied AI, where these models should be easily accessible, dynamically trained, and peer-reviewed not only by researchers but also by healthcare professionals.

In the medical research field, data sharing has always been challenging due to the presence of Protected Health Information (PHI), which should be carefully treated to preserve the confidentiality of individuals’ personal data. Despite these challenges, substantial efforts have been made by big institutions such as the World Health Organization (WHO) or the Centers for Disease Control and Prevention (CDC) to create public medical datasets, democratizing access for exploring research ideas. While these datasets are extremely valuable, they require meticulous de-identification and Institution Review Board (IRB) approval before they can be shared publicly, limiting the quantity, diversity, and modernity of the available datasets.

Regarding medical model development and sharing, significant efforts have also been made with frameworks such as MONAI [1], which stands for the Medical Open Network for AI. This is an open-source framework for deep learning in medical imaging. Initially developed by King’s College London and NVIDIA, it is now also supported by a huge community, with over 190 contributors from all over the world. MONAI provides a platform for building, training, and deploying deep learning models for medical image analysis. As an open-source project, it allows developers to both access and contribute to its codebase, fostering a collaborative environment for advancing AI technologies in healthcare. While this project marks a substantial step towards making medical AI models more accessible, it is important to note that the development and accessibility of these models are primarily driven by programmers rather than medical experts.

Another common challenge concerning model sharing is the model’s generalization problem. Despite big efforts to create platforms to host and give access to trained models, many of these models suffer from biases, methodological flaws, and a lack of reproducibility. This issue was illustrated by a team at the University of Cambridge in 2021, where they studied more than 300 COVID-19 machine learning models described in scientific papers in 2020. The study found that none of these models were really suitable for detecting or diagnosing COVID-19 from standard medical imaging [2]. To address this challenge, an interesting solution for model and indirect data sharing has emerged, known as federated learning. This method leverages data diversity over multiple institutions to generate more universally applicable models. In federated learning, a model architecture is trained across multiple decentralized institutions, each holding their local data samples. By sharing only the trained model weights without the necessity of sharing the raw data itself, this training technique enables the development of more generalizable models [3], preserving data privacy, which is especially important in the healthcare field. NVIDIA, along with its computational framework for model development, Clara, has made significant strides in creating such models. For example, a collaborative effort involving 20 different institutions globally used local data to train a predictive model for COVID-19 outcomes, which was then integrated using NVIDIA Clara’s framework [4]. This innovative approach solves the challenges associated with the complexities of sharing medical data and contributes to the creation of more generalizable models. 

In this context, we introduce the Discovery Viewer (DV) platform, a web application developed at the Biomedical Engineering and Imaging Institute at Mount Sinai (BMEII). DV started as an effort to create a collaboration bridge between healthcare professionals and researchers by providing a user-friendly environment to build dynamical medical models based on real medical needs. By leveraging the expertise of both groups, DV facilitates the development of cutting-edge medical AI models that remain accessible and applicable beyond the confines of academic publication. The platform also ensures secure, zero-footprint, web-based access, allowing medical experts to effortlessly review and annotate anonymized medical data. 

For AI model training, across all AI experience levels, DV is bundled with pre-trained deep learning models, enabling users to easily explore and test their model ideas through a transfer learning approach. While large and diverse datasets are crucial for training advanced pre-trained models, transfer learning has become the best approach for optimal model performance when working with limited datasets. For medical imaging, the platform provides RadImageNet (RIN) pre-trained models, a collection of pre-trained model architectures that were trained using a large multi-modality radiologic dataset of 1.35 million annotated medical images, including CT, MRI, and Ultrasound [5]. RIN pre-trained models have emerged as a leading option that provide high performance in numerous clinical tasks. It facilitates model development but also encourages collaborations, peer testing, and continuous model refinement, allowing dynamical model updates that are especially important in the medical field, where diseases and scanning techniques evolve rapidly.

Unlike many frameworks that are limited to a single data type, DV was designed in a modular way, based on the integration of different medical data viewers. Currently, DV serves as a dynamic platform, enabling the visualization, annotation, labeling, and scoring of medical images, physiological signals, and medical reports. But the platform’s design allows the future integration of other medical viewers, such as genomics or electronic health records (EHRs). By integrating diverse medical data types, DV facilitates the development of multimodal models, which present a significant advantage in performance over their single modality counterparts [6,7,8]. Furthermore, it offers the potential to serve as a digital twin platform through the intelligent integration of multiple medical data types and AI models. 

Numerous medical platforms are available in the market, but a considerable portion of them belong to private companies, limiting their accessibility. In Table 1, we explore alternative free solutions and evaluate the competitive advantages of DV.

DV also functions as a hub for dynamic AI medical models, storing essential details for each model, including training parameters, imaging modality, image input size, etc. This helps model organization purposes and version control. Additionally, DV promotes collaborations across institutions through the implementation of a federated learning approach, enabling the development of more generalizable models. As previously discussed, the shareability of these models facilitates indirect data sharing by just providing learned weights, PHI-free, that encapsulate the relevant data knowledge in terms of predictions. While this solution addresses the challenge associated with medical data sharing, we acknowledge that there could be specific scenarios where users may want to share cohorts. For this, DV facilitates the shareability of anonymized cohorts, keeping the data centralized avoiding duplications.

This paper showcases the outcomes of three distinct musculoskeletal AI models for regression, classification, and segmentation, respectively. These models were developed by three non-AI experts (a high-school intern at BMEII, an undergraduate engineering intern at BMEII, and a radiology resident at Mount Sinai) using the “Training Module” within DV’s platform. Each participant annotated their datasets and trained their respective models within the platform. Subsequently, the model’s performance was evaluated using a hold-off testing dataset.

## 2. Research Projects Using DV

### 2.1. Leveraging DV for Visualization and Annotation

The DV platform has proved to be a valuable tool in several research projects beyond just AI model development, resulting in multiple publications in peer-reviewed journals. Here, we present two projects conducted through the platform, illustrating its efficacy in expediting collaborative research initiatives. A summary of the DV projects setup is presented in Table 2.

#### 2.1.1. Spiral VIBE UTE-MRI for Lung Imaging in Post-COVID-19 Patients

Lung MRI screening has been challenging due to its short relaxation times; therefore, computed tomography (CT) has been the current gold standard for lung structure imaging but requires radiation exposure. Ultrashort echo time (UTE) MRI has emerged as a potential alternative for lung imaging without radiation. Additionally, the combination of UTE acquisitions with stack-of-spirals trajectories can improve imaging efficiency by making the scan time shorter so it can be obtained during a single breath hold. Spiral VIBE UTE-MRI combines ultra-short echo time acquisitions with a stack-of-spirals trajectory for imaging the lungs in a single breath hold. Compared to CT, the Spiral VIBE UTE-MRI is a radiation-free imaging technique that could be used for longitudinal lung studies, for example, recurrent controls of COVID-19-recovered patients (post-COVID-19 patients). The goal of this study was to analyze the motion sensitivity of different reordering schemes and breath-holding positions in Spiral UTE MRI to determine the optimized protocol for lung imaging. The diagnostic image quality for the optimized Spiral UTE-MRI protocol was then compared to CT in a group of post-COVID-19 patients [9].

For this study, DV served as the host for the Spiral UTE-MRI and CT images for a group of 36 subjects. It also provided a scoring metric tool for a group of three expert radiologists to easily access and evaluate six different lung image quality measurements. These measurements included scores for large arteries, large airways, segmental arteries, segmental broncho vascular structures, subsegmental vessels, and artifact levels (see Figure 1).

The results of the study indicated that Spiral VIBE UTE images acquired during the inspiratory breath-holding position were significantly better (*p* < 0.05) than other spiral imaging schemes. This radiation-free method takes advantage of the ultrashort echo time and the stack-of-spiral trajectories, achieving good image quality for lung assessment. It holds potential for longitudinal studies of lung structure changes, which might be important for certain patient populations, such as COVID-19-recovered patients (post-COVID-19 patients). 

#### 2.1.2. Automated Measurements of Leg Length on Radiographs by Deep Learning

Leg length discrepancy (LLD) is an orthopedic pathology defined by a difference in the total lower extremity length of greater than one centimeter. The diagnosis of LLD can be especially important for pre-operative evaluation for knee and hip arthroplasties. The determination of leg length requires multiple manual measurements on either a full-leg radiograph or computed tomography (CT) scanogram. The measurements are somewhat laborious, and the process can be automated through deep learning algorithms. The goal of this study was to develop a deep learning tool to automatically measure leg lengths from CT radiographs performed for pre-operative evaluation of knee arthroplasties [10].

In this study, DV was used to measure the leg length through the identification of three anatomic points (femoral head, medial femoral condyle/tibial plateau, and tibial plafond) for each leg. A total of 441 patients were included in this study, and all images were hosted on a DV project where they were annotated by an experienced muskuloskeletal radiologist (Figure 2).

The annotations were extracted from the platform in the form of a CSV file to develop the predictive models. The best model obtained a mean absolute error (MAE) of 17.64 pixels. Subsequently, DV was used to compare the model accuracy to the manual human expert measurements. A paired *t*-test was used to compare both groups obtaining a *p*-value higher than 0.05, indicating no significant differences between the groups.

### 2.2. Leveraging DV for Non-AI Expert Model Development

DV provides a training module with a comprehensive set of tools for building various AI models from scratch. To achieve this, it links specific project annotations to diverse pre-trained models and computational resources like GPUs and storage. This approach facilitates the development of AI models through the platform, which is especially important for individuals lacking expertise in AI. Here, we present different projects undertaken by three non-AI experts: a high-school intern at BMEII, an undergraduate engineering intern at BMEII, and a radiology resident at Mount Sinai. All three users, despite self-reporting their lack of expertise in AI model development, were assigned different muskuloskeletal tasks involving segmentation, regression, and classification. This was done to demonstrate DV’s capacity to seamlessly create and deploy AI models, even for those without AI expertise.

#### 2.2.1. Segmentation Task: Intravertebral Disk Segmentation on T1-Weighted Axial-View MRI Slices

Through the examination of various algorithms documented in the existing literature, it becomes evident that employing image segmentation models for boundary delineation is a widely utilized and efficient method in medical image analysis [11,12]. When it comes to spine imaging, radiologists frequently need to identify and segment different parts of the vertebrae. Manual segmentation can be laborious and time-consuming. Therefore, automatic segmentation algorithms have been introduced to efficiently segment and characterize the vertebrae. The applications of automatic segmentation models are diverse, spanning from fracture identification to the quantification of bone mineral density.

In this study, our volunteer was assigned to segment the Intervertebral Disk (IVD) on T1-weighted axial-view MRI slices of the lumbar spine (see Figure 3).

#### 2.2.2. Regression Task: Automated Measurement of Patellofemoral Anatomic Landmarks

In orthopedics, there is a clear clinical interest in further understanding the anatomy of the patellofemoral compartment. Inadequate characterization of this compartment can lead to frequent complications in knee arthroplasty for advanced osteoarthritis. Consequently, the identification of key anatomic landmarks, particularly those corresponding to the trochlear groove, plays a crucial role in understanding patellar biomechanics. This process involves the identification of seven key anatomical landmarks within the axial view of the patellofemoral compartment. These landmarks enable the measurement of various parameters that quantify morphological characteristics. 

For this project, our volunteer was responsible for annotating the seven points on axial CT images of the knee: three points along the anterior femoral trochlea at the medial, central, and lateral aspects, respectively; two points along the most peripheral medial epicondyle and most posterior medial condyle; and two points along the most peripheral lateral epicondyle and most posterior lateral condyle (see Figure 4). These points are in accordance with prior morphometric studies of the distal femur [13].

Recognizing the anatomical landmarks, as well as analyzing axes and angles, can help define morphological classifications. A primary goal in clinical applications is to enhance implant positioning, replicating the native trochlear anatomy and mechanism. This also includes optimizing the surgical approach and evaluating existing arthroplasty implants to identify areas where design improvements are needed.

#### 2.2.3. Classification Task: Normal vs. Osteoarthritis Knee Axial View CT Slice Classification

Osteoarthritis (OA) is a degenerative joint disease frequently affecting large joints such as the hips, shoulders, knees, and spine. Detecting OA in its early stages is essential, as it facilitates timely interventions to prevent cartilage degeneration and bone damage [14]. Consequently, reducing chronic pain and enhancing joint function. 

Identifying and addressing OA in its early phases not only holds the potential to alleviate discomfort and maintain joint function but also plays a crucial role in advancing the field of OA research and therapeutics. Such early interventions can significantly contribute to the development of new therapies to delay the need for joint replacement surgery.

This classification task involved a straightforward classification process, where axial CT scans were categorized into normal or osteoarthritic at the slice level (see Figure 5).

## 3. Materials and Methods

### 3.1. Web App Design

DV was designed and built using the Django web framework, with Python as the backend programming language and HTML, CSS, and JavaScript for the frontend components. The platform leverages the DWV JavaScript library to enable a web-based DICOM viewer and Chart.js to enable the web-based physiological signal viewer. Additionally, all annotations are performed using Konva.js. These annotations, along with labels and scores, are sent from the client side to the main server, where they are securely stored in a local SQL database. A schematic of the overall platform architecture can be observed in Figure 6.

The image viewer, as depicted in Figure 7A, is structured based on cases, each of which can contain multiple scans. It offers functionalities such as scrolling, zooming, and windowing to visualize the images. For annotation purposes, the viewer provides tools for creating landmarks, regions of interest (ROIs), bounding boxes, and lines. Additionally, it offers tools for scoring and labeling, allowing reviewers to assign custom assessment metrics and categories either at the scan or slice level. 

The signal viewer, depicted in Figure 7B, is structured similarly to the image viewer—it is organized by case and recordings, with each recording containing one or multiple channels that could represent various physiological signals (e.g., ECG, Resp, EDA, etc.). This viewer allows users to scroll through the signal, zoom in and out, and annotate specific time events. Additionally, it also offers labeling and scoring tools as needed.

The report viewer, depicted in Figure 7C, follows a similar structure to the previous viewers. It is organized by case, with each case containing one or multiple reports. Users can annotate specific sections of the report or apply custom scores or tags to the entire report.

DV’s platform is specifically designed to host projects, models, and cohorts of each user. The cohorts are easily created by the integration of multiple databases with different medical data types like the Imaging Research Warehouse (IRW), a massive image database developed at BMEII that hosts clinical imaging with electronic health records of more than 1 million Mount Sinai patients. With the correct permission, these cohorts can be placed in DV studies aiming to develop medical AI models or to review the medical data. The creator can control reviewers’ access, determine the shareability of annotations, check reviewers’ progress, download the annotated data, and add questionnaires, scoring metrics, and labels at will. The users also have the option to create studies with artificial datasets, PHI-free, generated using generative models. For this, DV is bundled with RadImageGAN (RIGAN), a set of generative models for the creation of synthetic medical data (see Figure 8). This is particularly beneficial to avoid extended waiting periods and to start working on project ideas while waiting for the Institutional Review Board (IRB) approval to access the actual data. 

For model development, DV is bundled with RadImageNet (RIN) pre-trained models to effectively apply transfer learning and obtain performance results on limited datasets. For non-AI experts, DV offers a user-friendly training module (Figure 9A) where non-programmers can easily explore their predictive model ideas. The platform is connected to an NVIDIA DGX server with 8x A100 GPUs, enabling the utilization of graphic units for faster training. The models can be developed at the slice level, 2D, at the scan level, 3D, or even at the subject level to be able to develop multi-modal models. Post-training, the platform automatically integrates the developed model into its medical model hub, granting users immediate access for testing purposes. The training module also keeps a version control record of the developed models, preserving detailed information about the training and tuning datasets, the architecture employed, and more (see Figure 9B).

### 3.2. AI Project Setup

Three separate AI projects were conducted by individuals lacking AI expertise: a high-school intern at BMEII, an engineering undergrad intern at BMEII, and a radiology resident at Mount Sinai. All three participants self-reported non-expertise in developing AI models, and they were assigned three distinct musculoskeletal tasks involving segmentation, regression, or classification. This was done to demonstrate DV’s ability to enable the seamless development and deployment of AI models, even for non-experts.

Only 80% of the data was given to each volunteer; the remaining 20% was held off for testing. Also, a small proportion of this training dataset was annotated to provide guidance in terms of the expected annotations. Besides this project setup, a set of straightforward instructions were provided to the non-expert developers, including the following elements:-A tutorial on how to use the annotation tool (windowing, scrolling, zooming, and annotation/labeling).-A tutorial on how to use the training module to select the annotated dataset and train a new model through a transfer learning approach. The options for model training are limited to just a couple of parameters: batch size, number of epochs, learning rate, and selection of a pre-trained model. Regarding RIN pre-trained models, DV presently offers the following options. However, the plan is to expand and enhance the selection of pre-trained networks in line with the evolving architectural trends in the literature.
○For segmentation: ResNet50UNet. ○For regression: Resnet50, DenseNet121, InceptionV3, and InceptionResNetV2. ○For classification: Resnet50, DenseNet121, InceptionV3, and InceptionResNetV2.

RadImageNet pre-trained architectures of Inception-ResNet-v2, ResNet50, DenseNet121, and InceptionV3 were trained from scratch, initiating the training process with randomly initiated weights. All 1.35 M images were uniformly resized to a resolution of 224 × 224 pixels before being fed into the neural networks. Subsequent to the convolutional layers, a global average pooling layer, a dropout layer with a 50% rate, and a softmax-activated output layer were integrated. These models were designed to output a probability distribution across the 165 labels in the RadImageNet dataset, indicating the likelihood of each image belonging to a specific label.

The volunteers had two weeks to annotate the datasets, choose an RIN-pre-trained architecture, and train the predictive models. After this period, they reported their best models based on the validation performance metrics given by the platform.

### 3.3. Datasets

#### 3.3.1. Knee CT Dataset

This dataset was obtained from a prior study titled “Deep Learning for Automated Measurement of Patellofemoral Anatomic Landmarks” conducted by Zelong Lui et al. [15]. It comprises axial CT images of the knee, which were gathered from six distinct locations within the Mount Sinai Health System from April 2017 to May 2022. These studies were conducted using CT systems from four different manufacturers (GE, Siemens, Toshiba, and Siemens) and featured a slice thickness spanning from 0.625 to 5.0 mm. The cohort encompasses a total of 483 patients, with 206 being healthy individuals and 277 individuals diagnosed with osteoarthritis. Additionally, the ground truth of seven trochlear groove anatomic landmarks at the axial slice level were annotated by radiology and orthopedic residents at Mount Sinai.

For the regression task, we selected 80 scans from the pool of healthy subjects. Out of these, 61 scans were included in the training dataset, while the remaining 19 were reserved for testing. Additionally, 10 scans with ground truth were available to the volunteer, offering guidance on the expected annotations. 

For the classification task, a total of 156 scans were extracted, encompassing subjects with both healthy and osteoarthritic conditions. Among these scans, 124 were included in the training dataset, comprising 64 scans from abnormal knees and 60 from normal ones. For testing purposes, 32 scans were set aside, evenly split between 16 normal knees and 16 abnormal ones.

#### 3.3.2. Lumbar Spine MRI Dataset

This is an open-source dataset available on the Mendeley Data website, which is a free and secure cloud-based repository to store, share, and access data. It comprises anonymized clinical MRI studies from 515 patients experiencing symptomatic back pains [16]. The dataset includes ground truth labels for axial view slices of the lumbar spine, focusing on four specific labeled regions of interest: (1) Intervertebral Disc (IVD), (2) Posterior Element (PE), (3) Thecal Sac (TS), and (4) the Area between Anterior and Posterior (AAP) vertebrae elements. The labeling procedure was conducted by five individuals under the supervision of an expert radiologist, utilizing T1-weighted axial-view MRI slices from the last three IVDs. The slices in this dataset have a resolution of 320 × 320 pixels, with a precision resolution of 12 bits per pixel. The slice thickness is 4 mm, and the xy resolution is 0.6875 × 0.6875.

For the segmentation task, a total of 2472 single-slice axial view images were utilized for the training dataset, and 618 slices were reserved for testing. The ground truth was provided for 100 slices to guide the annotation process.

### 3.4. Model Building

#### 3.4.1. Best Model for Segmentation Task

The radiology resident developed this model, annotating a total of 1166 single-slice scans from the training dataset. The best model was achieved by training an RIN-pretrained ResNet50 UNet for over 200 epochs and utilizing a batch size of 16. The training samples were split into 80% for training and 20% for tuning. For handling segmentation problems, the platform employs bce_jaccard_loss as the loss function and iou_score as the metrics, sourced from the Python segmentation_models module [17]. The optimizer used was Adam, and the learning rate was set at 0.001. The model input target size is (224 × 224), requiring images to be normalized and resized before being fed into the architecture.

To ensure proper normalization, the platform uses Contrast Limited Adaptive Histogram Equalization (CLAHE), a variation of adaptive histogram equalization that prevent contrast over-amplification. The platform uses OpenCV’s CLAHE class [18] with the following parameters: clipLimit = 2 and tileGridSize = (8, 8). 

#### 3.4.2. Best Model for Regression Task

The undergrad BMEII intern developed this model. In addition to the 10 scans with provided annotations, the intern annotated the remaining 51 scans at the slice level, completing the dataset of 61 scans used for training. For model building, our developer employed an RIN-pretrained InceptionV3 architecture, training the model at the slice level for 200 epochs. The training process utilized a batch size of 16 and a learning rate of 0.001, with an 80% training and 20% tuning data split.

For regression problems, DV employs the mean absolute error (MAE) as the loss function and metric. Like segmentation problems, the optimizer used is Adam, the image normalization method applied is CLAHE, and the input target size for the model is 224 × 224.

#### 3.4.3. Best Model for Classification Task

The high-school BMEII intern was responsible for developing this model, categorizing 124 scans into “healthy” or “osteoarthritic” knees. To develop the model, the intern used an RIN-pretrained ResNet50 architecture and conducted training at the slice level over 100 epochs, with a batch size of 16 and a learning rate of 0.001. The training dataset was divided into 80% for training and 20% for tuning.

For classification problems, DV employs binary cross-entropy as the loss function and classic accuracy as the chosen metric. Similarly, as for segmentation and regression tasks, the optimizer used is Adam, the image normalization method is CLAHE, and the input target size of the model is 224 × 224. 

## 4. Results

### 4.1. Segmentation Model Performance

To test the performance of the best segmentation model, we used the following metrics: jaccard score, dice score, sensitivity, and specificity. The average and standard deviation results for all test samples can be observed in Table 3. 

### 4.2. Regression Model Performance

The best regression model was tested against the hold-off testing dataset, obtaining a mean absolute error of 14.27 pixels. In comparison to the leg length project described earlier, which achieved a mean absolute error of 17.64 pixels, we can regard this model as a promising initial step for automating landmark location. The Bland–Altman plot for the X and Y coordinates of all predicted points vs. the ground truth can be observed in Figure 10.

### 4.3. Classification Model Performance

For the classification model, the best model obtained an accuracy of 0.94, a sensitivity of 0.92, and a specificity of 1 on the testing dataset. The ROC-AUC can be observed in Figure 11.

## 5. Discussion and Conclusions

The high performance exhibited by the three models on the testing datasets shows the effectiveness of transfer learning through RIN pre-trained models in facilitating efficient model training within DV’s platform. This approach opens the door to infinite medical applications envisioned not only by researchers but also by healthcare professionals, who can now easily develop and use their own model ideas. By providing a space for non-AI experts to independently develop their model concepts, we aim to narrow the gap between healthcare professionals and this new technology, thus advancing the integration of AI models into clinical environments. From diagnostic supports to the automation of anatomical measurements, these models can be tailored to their specific medical needs. Furthermore, it creates a unique educational environment for these non-AI experts to understand the different steps in the process of developing medical AI models. 

Throughout the paper, we have showcased various examples of medical research projects that benefited from DV’s platform to host the data, annotate it, develop a variety of medical models, and deploy these models for testing purposes. Notably, the platform modular design allows the seamless integration of models developed in various environments other than DV, fostering collaboration among researchers. Recognizing the potential biases inherent in some models, we emphasize the importance of transparency, allowing users to access, test, provide feedback, and engage in refining these shared models through a fine-tuning approach or a federated learning process. 

Overall, the Discovery Viewer platform offers a collaborative environment for healthcare professionals and researchers to work together towards the development and deployment of cutting-edge AI models in medical research. The platform’s features, including zero-footprint and secure web-based access, support for multiple data types, and seamless development and deployment of models, make it an attractive option for medical professionals and researchers looking to collaborate and drive innovation.

## 6. Future Work

While the platform is currently operating within the confines of the Mount Sinai network, we envision, in the future, potentially hosting it in the cloud, extending access to various institutions. This goal aligns with our vision of adopting a federated learning approach, paving the way to generate more generalizable models. This democratization of advanced AI models holds the promise of enhancing medical assessments and manual processes, thereby revolutionizing the healthcare landscape.

An additional avenue of development for the platform includes the incorporation of additional medical data viewers, such as genomics or electronic health records (EHR). By integrating these new viewers, we would also look to collaborate with other Mount Sinai data platforms, such as AIR.MS [19], which is a cloud-based, multi-modal medical data platform that can give researchers access to a vast dataset. This way, DV sets out to become a medical viewer research platform to host, display, and annotate/review multi-modal cohorts. The concept of “discovery viewer” comes from the idea that users are capable of visualizing not only one type of data at a time but also to simultaneously use multiple medical data viewers, discovering the whole universe of the patient’s health status.

DV’s foundation on medical data viewers and medical AI models sets the stage for a more comprehensive and varied healthcare platform, potentially serving as the foundational platform of digital twins.

## Figures and Tables

**Figure 1 bioengineering-10-01396-f001:**
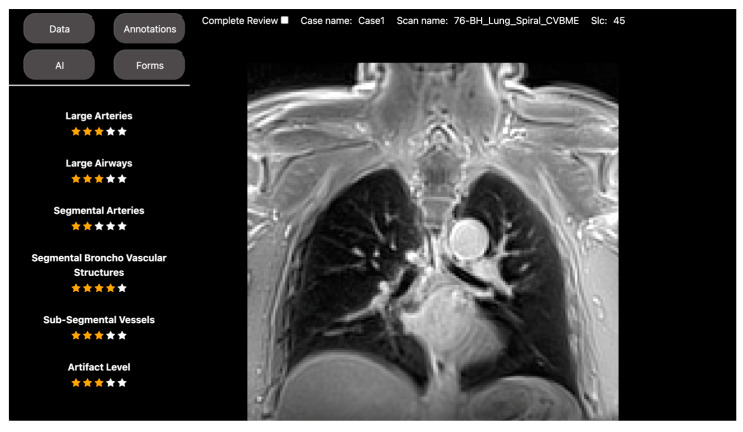
DV’s scoring tool for image quality assessment of Spiral UTE MRI imaging. This figure depicts DV’s visual display of images obtained using the Spiral UTE MRI sequence alongside six different image quality metrics. This study incorporates the following quality scoring metrics: artifact levels, sub-segmental vessels, segmental broncho vascular structures, segmental arteries, large airways, and large arteries. All the results were locally saved in an SQL database and subsequently used for statistical tests to compare the image quality with the standard chest CT scan.

**Figure 2 bioengineering-10-01396-f002:**
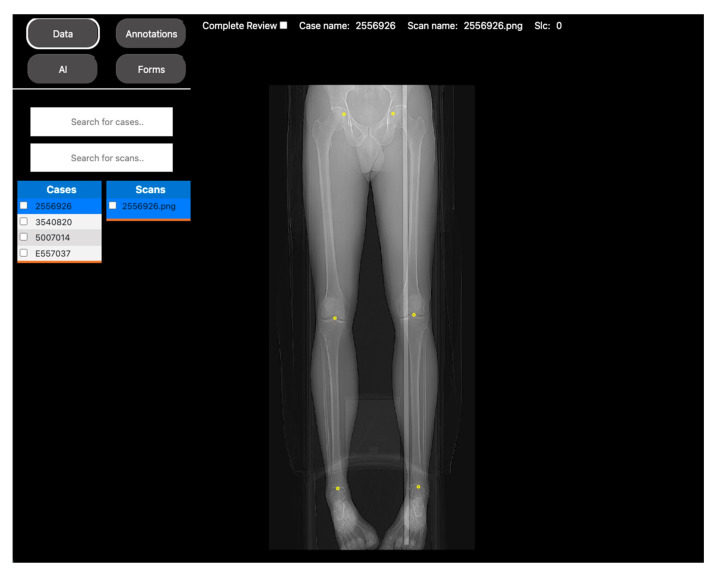
DV’s landmark annotation tool for leg length anatomic measurements. This figure depicts DV’s visual display of a full-length radiograph for landmark anatomical annotations. In this project, an expert radiologist marked three anatomical points for each leg: one at the femoral head, another at the medial femoral condyle plateau, and a third at the tibial plafond. The annotations were securely stored in an SQL database and later used to train a regression model to automate the manual process.

**Figure 3 bioengineering-10-01396-f003:**
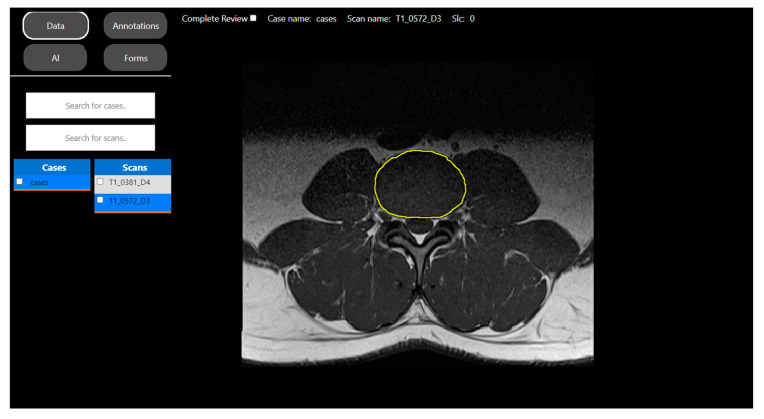
DV’s annotation tool for region of interest segmentation. This image corresponds to DV’s visual representation of a T1-weighted axial-view MRI slice used for IVD segmentation (yellow ROI in the image). These ROIs are securely stored and serve as the foundation for the development of a segmentation model to automatically segment the anatomical region of interest at the slice level.

**Figure 4 bioengineering-10-01396-f004:**
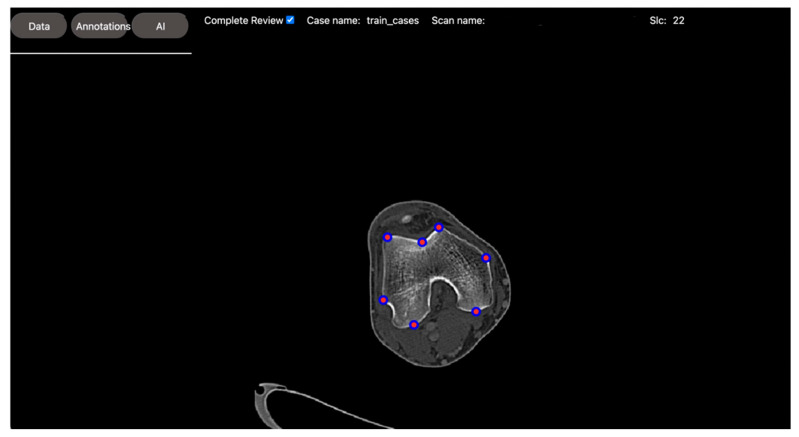
DV’s annotation tool for anatomic landmark annotations. The image shows DV’s visual presentation of an axial CT scan of the knee. Seven anatomical points were placed at the slice level in accordance with prior morphometric studies. These annotations were stored and used to develop regression models to automatically identify these points and classify different knee morphologies for bio-mechanical assessments.

**Figure 5 bioengineering-10-01396-f005:**
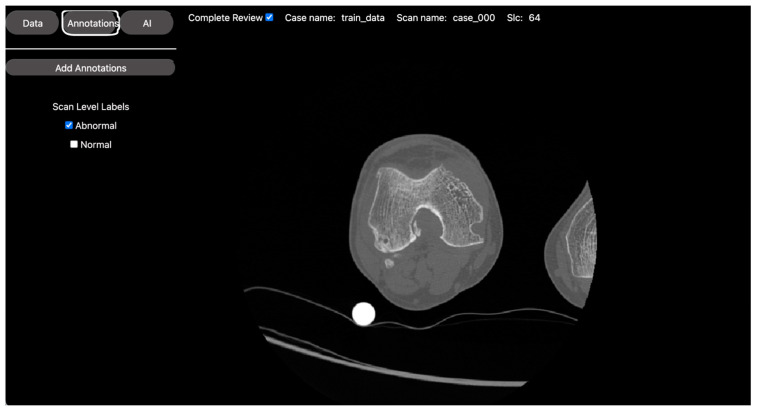
DV’s labeling tool. This image corresponds to DV’s visual presentation of an axial CT scan of an osteoarthritic knee. The platform allows the project owner to set custom labels, which are assigned by the expert medical reviewers. Thus, the platform facilitates the rapid establishment of a ground truth for a set of images, enabling the training of classification models.

**Figure 6 bioengineering-10-01396-f006:**
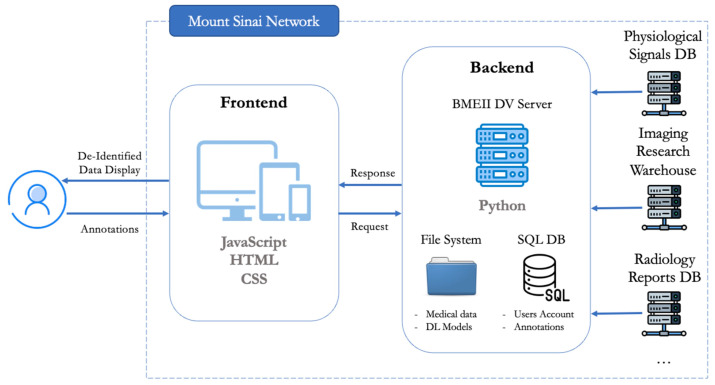
Platform architecture scheme. The current platform operates within the Mount Sinai network. The backend is connected to various medical databases, simplifying the creation of cohorts. Data are organized and stored in a file system per project under the home space of the creator. The backend is also responsible for user account management, ensuring projects/cohorts/models access to only users with the right permissions. On the frontend, only anonymized data are displayed, and reviewers can interact with the viewers to annotate the medical data. The annotations are then securely stored in SQL databases on the backend server.

**Figure 7 bioengineering-10-01396-f007:**
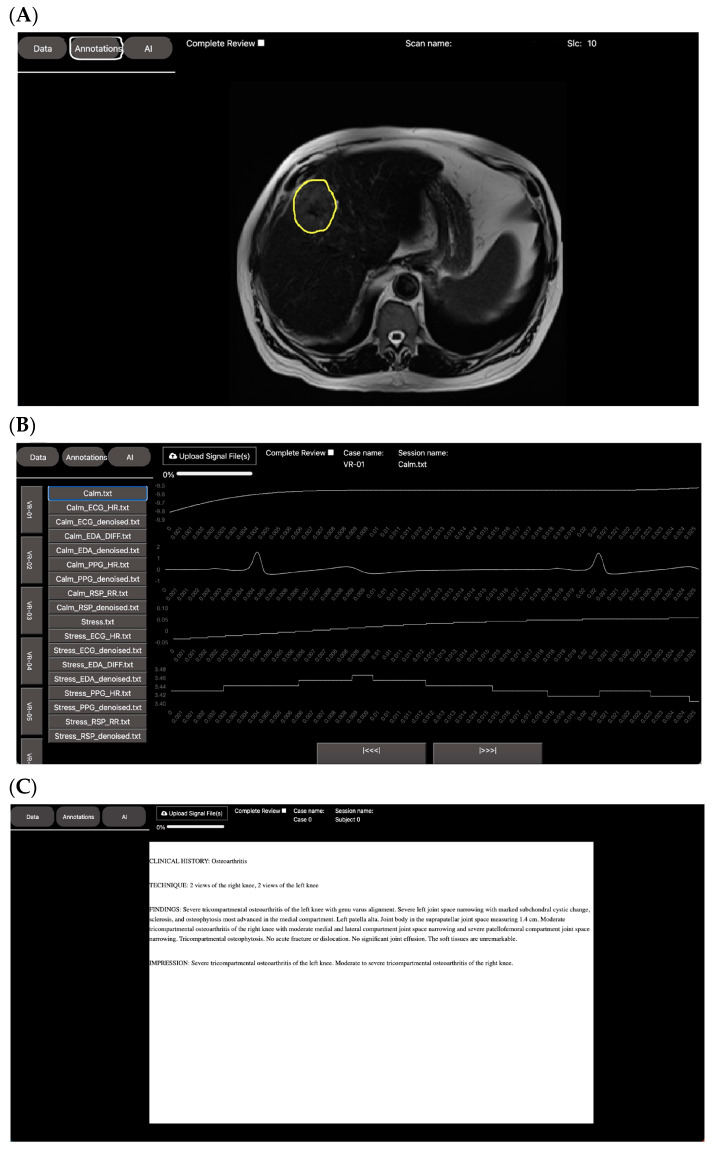
Different medical data viewers integrated in DV. (**A**) DICOM viewer; (**B**) physiological signal viewer; (**C**) medical report viewer. Various medical data types require different viewers for data exploration and annotation. DV’s platform was designed in a modular way to enable the integration of multiple medical viewers. This integration facilitates the combination of medical data to create multi-modal ensemble models.

**Figure 8 bioengineering-10-01396-f008:**
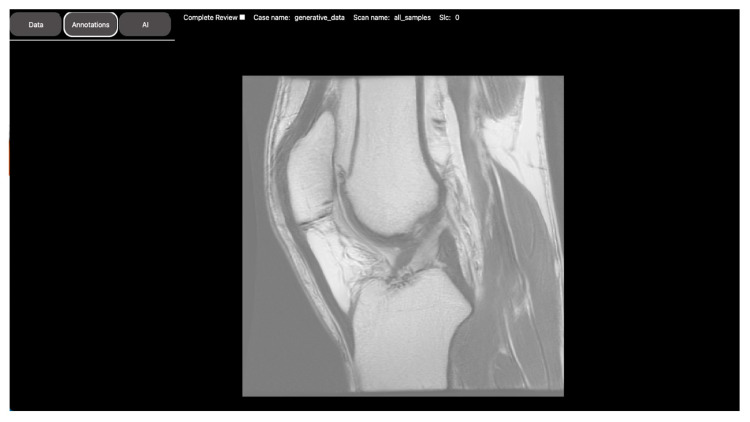
An artificially generated sagittal image of the knee produced using RadImageGAN (RIGAN) generative models. RIGAN is a multi-modal radiologic data generator, developed by training StyleGAN-XL on RadImageNet. It can generate high-resolution synthetic medical imaging datasets across 12 anatomical regions and 130 pathological classes in 3 modalities.

**Figure 9 bioengineering-10-01396-f009:**
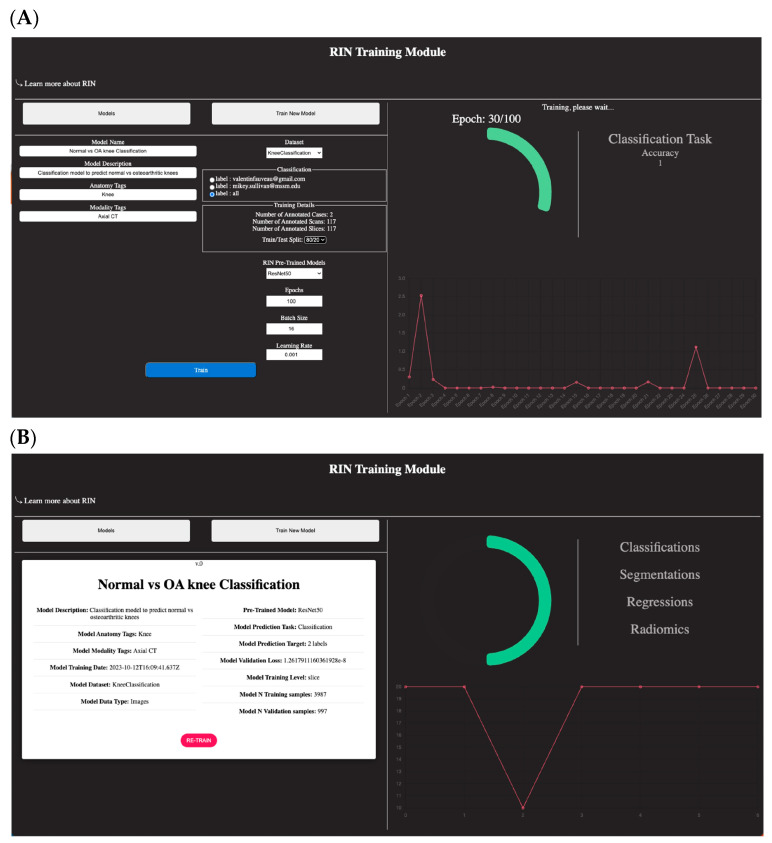
DV’s training module: (**A**) training module interface; (**B**) model details sheet. DV’s platform is bundled with RadImageNet pre-trained networks, enabling users to train models through a transfer learning approach. The platform automatically recognizes the type of annotations present in each project, offering a list of available pre-trained networks for efficient transfer learning. This feature provides a user-friendly space for non-AI experts to easily develop AI models for various medical needs.

**Figure 10 bioengineering-10-01396-f010:**
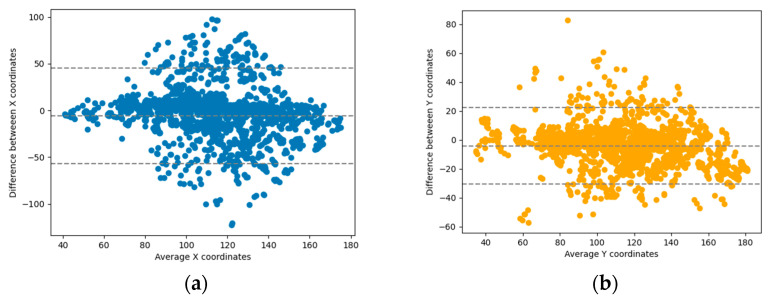
Bland–Altman plot of the X (**a**) and Y (**b**) coordinates of the regression model prediction vs. ground truth. The regression task focused on accurately positioning 7 anatomic landmarks in the axial MRI view of the knee. In this task, 61 scans were utilized for training the model, while 19 scans were set aside for testing purposes. The model was trained at the slice level and achieved a mean absolute error of 14.27.

**Figure 11 bioengineering-10-01396-f011:**
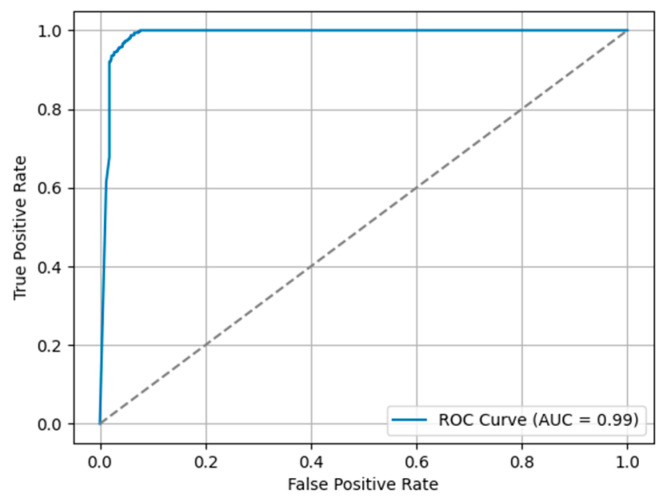
ROC-AUC curve of the knee classification task. The classification task involved the prediction of healthy and osteoarthritis (OA) knee scans based on axial-oriented MRI images at the slice level. The training dataset included 124 scans, with 64 OA knees and 60 healthy ones. The model’s performance was evaluated on a separate testing dataset, which included 32 scans evenly divided between 16 healthy and 16 OA knees. The model obtained an AUC of 0.99, an accuracy of 0.94, a sensitivity of 0.92, and a specificity of 1.

**Table 1 bioengineering-10-01396-t001:** Exploration of free medical research platforms and DV’s competitive advantage. 1. Web-based solution: This key feature allows zero-footprint accessibility, enabling centralized data management, project access control, and streamlined tracking of the annotation/review process. 2. Multi-modality medical data viewers: This unique feature is significantly relevant since the integration of multiple medical data types has proven to offer a considerably enhanced understanding of a patient’s healthcare status. Therefore, for medical model development, the integration of multimodal AI models is paramount to achieve the highest model performance. 3. AI model training: DV smartly leverages pre-trained networks, allowing users to easily train AI model ideas by linking project annotations with DL architectures. This establishes a user-friendly environment, particularly beneficial for non-AI experts to develop their model concepts. While other platforms like MONAI also offer a framework for AI model development, it requires programming experience, limiting the accessibility of this framework primarily to researchers. 4. AI model deployment: DV streamlines the deployment process by automatically deploying all models developed within the platform, facilitating direct testing by users. Additionally, it supports the integration of models developed in external environments by just incorporating a Docker containerized model. Other platforms such as MONAI, 3D Slicer, and OHIF also support model integration, but the process is not as straightforward and demands some programming expertise. 5. Accessibility: As discussed in point 1, being web-based offers a substantial advantage in providing access to all stakeholders and fostering collaborative environments between researchers and healthcare workers. Alternative platforms: MONAI: https://monai.io/ (accessed on 30 November 2023); 3DSlicer: https://www.slicer.org/ (accessed on 30 November 2023); OHIF: https://ohif.org/ (accessed on 30 November 2023).

	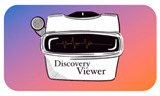	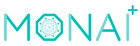	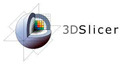	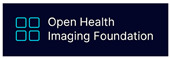
**Web Based**	**✔**	**✔**	ⅹ	**✔**
**Medical multi-modality data viewers**	**✔**	ⅹ	ⅹ	ⅹ
**AI model training**	**✔**	**✔** *(Requires programming)*	ⅹ	ⅹ
**AI model deployment**	**✔**	**✔** *(Requires programming)*	**✔** *(Requires programming)*	**✔** *(Requires programming)*
**Accessibility**	*Zero-footprint and secured web-based access*	*Requires Python based installation packages*	*Desktop software*	*Zero-footprint web-app*

**Table 2 bioengineering-10-01396-t002:** DV project setup overview.

Project Name	Spiral VIBE UTE-MRI for Lung Imaging in Post-COVID Patients [9]	Automated Measurements of Leg Length on Radiographs by Deep Learning [10]
Objective	Assess the diagnostic image quality of a novel MRI sequence to image the lungs (see Figure 1).	Develop a DL model to automatically identify anatomical landmarks in full leg radiographs (see Figure 2).
Methodology	Three radiologists assessed the diagnostic image quality of a group of post-COVID-19 patients who underwent both an MRI and a CT scan of the lungs.	A radiologist annotated three anatomical landmarks in a cohort of full-length radiographs.
DV project set up	Each patient’s MRI and CT images, along with the diagnostic quality metrics for scoring, were incorporated into a DV project. Access to the project was then provided to the three radiologists to review the images.	The training set of the cohort was located into a DV project, along with the points annotation tool. Access was given to the radiologist to annotate the images.
Results	The scores were extracted from the platform as a CSV file for statistical analysis.	The coordinates of the annotations were extracted as a CSV file to use as the ground truth for DL model training.

**Table 3 bioengineering-10-01396-t003:** Segmentation model performance. The segmentation task consisted in segmenting the Intervertebral Disk (IVD) on T1-weighted axial-view MRI slices of the lumbar spine. A total of 1166 single-slice axial view images were utilized for training and 618 slices were reserved for testing. The performance on the testing dataset is detailed in the table.

	Jaccard Score	Dice Score	Sensitivity	Specificity
Mean	0.89	0.93	0.9	0.99
Std	0.14	0.14	0.14	0.001

## Data Availability

We are currently running a proof of concept internally at Mount Sinai. In the future, we hope to host this platform as a service in the cloud to make it available to the public and exploit the benefits of having a higher collaboration community working on updating and peer reviewing the predictive medical models. If you want to have an early access to this platform please contact Valentin Fauveau (valentin.fauveau@mssm.edu).

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
