# Peer review of "Discovery Viewer (DV): Web-Based Medical AI Model Development Platform and Deployment Hub"

_bioengineering, 2023, doi:10.3390/bioengineering10121396_

Round 1

Reviewer 1 Report

Comments and Suggestions for Authors

The paper introduces Discovery Viewer (DV), a web-based medical AI platform. However, some revisions should be made to improve the manuscript:

  1. The abstract needs to be shorter and more concise.
  2. The research objectives should be stated more clearly, outlining the specific goals of the study. For instance, the authors could provide a more detailed analysis of other medical AI platforms and state the contributions and advantages of the proposed platform in the introduction section.
  3. The authors need to provide more information on model details and experimental results for all models in sections 3.2 and 3.4.
  4. A thorough language edit and spell-check should be conducted.
  5. There is a minor comment regarding line 313. It should be corrected to Figure 6 instead of Figure 4.
Comments on the Quality of English Language
  1. Minor editing of English language required

Reviewer 2 Report

Comments and Suggestions for Authors

1.In the paper the authors proposed the design of a special Web-Based Medical AI Model Development Platform and Deployment Hub (Discovery Viewer – DV). DV is targeted to creation of a collaborative environment between doctors and researchers. The system provides the visualization, annotation, labeling and scoring of the medical images, physiological signals or medical reports. The authors described the DV's ultimate goal as a special support system (a central hub) for dynamical AI medical models.

2.In the paper three distinct musculoskeletal AI models for regression, classification and segmentation are described.

3.The authors described their experiments in the implementation of the Web-Based Medical AI Model Development Platform and Deployment Hub (Discovery Viewer – DV). The system experiments were based on the works of three types of non-AI experts (a high school BMEII intern, an undergraduate BMEII intern, and a first-year radiology resident).

4.The experiments demonstrate the capability of DV to make medical AI models accessible to a wider audience. The experiments results show the advanced integration AI into clinical environments.

5. In the section 2 ‘Research projects using DV’ it may be useful for readers to present information on other research/design efforts at the general structured level (i.e., as a generalized table and/or figure/framework).

6. In the section 3 Materials and Methodit may be reasonable to present the authors efforts at the general structured level (i.e., as a generalized table and/or figure/framework).

7.The authors system, its usage, and the design process will have a real interest for many professionals. The authors work was realized in hospital.

8.It may be reasonable to extend the title of the section ‘Discussion’: ‘Discussion and conclusion’.

9.It may be useful to point out some other types of Web-based clinical decision support systems (with corresponding references). This addition will be important.

10.The section ‘Discussion’ may be extended by small part on future prospective works.

11.In general the author paper is very interesting and will be very useful for many readers. The paper material is presented at a very good level. The paper can be accepted (after minor revision) .

Reviewer 3 Report

Comments and Suggestions for Authors

Summary:
The manuscript presents an engaging study on the development of a web-based medical AI model platform, designed to be accessible for non-AI experts. This platform enables users to train AI models for specific tasks. While the concept is intriguing, there are several areas in the study that require further clarification or revision.

Major Issues:

  1. Data Sharing Discussion (Introduction): The manuscript extensively discusses "data sharing" in the initial sections. However, the relevance of this discussion to the proposed model platform is not clearly established. Could the authors elucidate how "data sharing" benefits or integrates with their platform and consider restructuring parts of the introduction to clarify this connection?
  2. Model Accessibility (General Concern): The emphasis on "data sharing" seems at odds with the fact that the model itself is not publicly accessible, not even for review purposes. This raises questions about the practical utility and verifiability of the model. Can the authors address this discrepancy?
  3. Annotations and Predictive Models (Section 2.1.2): The manuscript states that annotations were extracted as CSV files to develop predictive models. If the leg length can be directly determined from the annotated points, the necessity of a predictive model in this context is unclear. Could the authors provide a rationale for this approach?
  4. Duplication of Content (Section 2.2 and Results): It appears that Section 2.2 has been duplicated in the results section. This redundancy should be addressed by reorganizing these sections for clarity and coherence.
  5. Figure Quality and Organization: There is a significant need to improve the quality of all figures, except those in the results section. The current screenshots of the platform are not sufficiently informative, exhibit large black margins, and the text and images are difficult to discern. Furthermore, the figure numbering is inconsistent. I recommend a thorough revision of these figures to ensure they are publication-ready.
Comments on the Quality of English Language

The quality of the English Language is ok for publish.

Round 2

Reviewer 3 Report

Comments and Suggestions for Authors

The authors answered all the issues and the new version looks good to me.